# Incretin Hormones and Type 2 Diabetes—Mechanistic Insights and Therapeutic Approaches

**DOI:** 10.3390/biology9120473

**Published:** 2020-12-16

**Authors:** Geke Aline Boer, Jens Juul Holst

**Affiliations:** 1Department of Biomedical Sciences, Faculty of Health and Medical Sciences, University of Copenhagen, 2200 Copenhagen N, Denmark; alineb@sund.ku.dk; 2NNF Center for Basic Metabolic Research, Faculty of Health and Medical Sciences, University of Copenhagen, 2200 Copenhagen N, Denmark

**Keywords:** GIP, GLP-1, incretins, T2DM

## Abstract

**Simple Summary:**

When we ingest a meal, our intestine secretes hormones that are released into the bloodstream. Amongst these hormones are the incretins hormones which stimulate the release of insulin from the pancreas which is essential for the regulation of in particular postprandial glucose concentrations. In patients with type 2 diabetes, the effect of the incretins is diminished. This is thought to contribute importantly to the pathophysiology of the disease. However, in pharmacological amounts, the incretins may still influence insulin secretion and metabolism. Much research has therefore been devoted to the development of incretin-based therapies for type 2 diabetes. These therapies include compounds that strongly resemble the incretins, hereby stimulating their effects as well as inhibitors of the enzymatic degradation of the hormones, thereby increasing the concentration of incretins in the blood. Both therapeutic approaches have been implemented successfully, but research is still ongoing aimed at the development of further optimized therapies.

**Abstract:**

Glucagon-like peptide-1 (GLP-1) and glucose-dependent insulinotropic polypeptide (GIP) are secreted from the gut upon nutrient stimulation and regulate postprandial metabolism. These hormones are known as classical incretin hormones and are responsible for a major part of postprandial insulin release. The incretin effect is severely reduced in patients with type 2 diabetes, but it was discovered that administration of GLP-1 agonists was capable of normalizing glucose control in these patients. Over the last decades, much research has been focused on the development of incretin-based therapies for type 2 diabetes. These therapies include incretin receptor agonists and inhibitors of the incretin-degrading enzyme dipeptidyl peptidase-4. Especially the development of diverse GLP-1 receptor agonists has shown immense success, whereas studies of GIP monotherapy in patients with type 2 diabetes have consistently been disappointing. Interestingly, both GIP-GLP-1 co-agonists and GIP receptor antagonists administered in combination with GLP-1R agonists appear to be efficient with respect to both weight loss and control of diabetes, although the molecular mechanisms behind these effects remain unknown. This review describes our current knowledge of the two incretin hormones and the development of incretin-based therapies for treatment of type 2 diabetes.

## 1. Introduction

Hormones secreted from the epithelium of the gastrointestinal tract, the incretin hormones, play an important role in normal glucose tolerance. By stimulating insulin secretion in a glucose-dependent manner, they ensure that postprandial glucose levels do not increase excessively. This effect, the incretin effect, is dose-dependent so that nearly the same postprandial glucose excursions are produced despite increasing carbohydrate contents of the meal [1]. The incretin effect is usually measured by comparing the insulin responses to oral and intravenous glucose administrations resulting in similar glucose excursions. It can also be estimated by comparing the amounts of glucose that must be infused intravenously to mimic the concentrations induced by the oral administration, and from such comparisons, it appears that the incretin effect may be responsible for up to five-fold increases in postprandial glucose clearance [1]. It turns out that the oral carbohydrate administration stimulates the secretion of the two gut hormones GIP (glucose-dependent insulinotropic polypeptide) and GLP-1 (glucagon-like peptide-1) which stimulate insulin secretion, and the stimulated insulin secretion, in turn, is responsible for the increased disposal of glucose [1,2]. The incretin secretion, and hence the insulin response is dependent on the ingested amount of carbohydrates, which explains the dependency of the incretin effect with the ingested amount. Recently, it has been possible to estimate the relative contributions of the two hormones to insulin secretion in response to the glucose load using potent and specific antagonists of the GIP and the GLP-1 receptors, namely exendin 9–39 for the GLP-1 receptor and GIP(3–30)NH_2_ for the GIP receptor [3]. Judging from the combined and separate insulin responses to the blockers (corrected for simultaneously occurring changes in plasma glucose concentrations), it could be calculated that, in response to a 50 g oral glucose load, GIP was responsible for 44% of the insulin response, GLP-1 for 22% and glucose alone for 33% of the response [4]. In 1986 it was demonstrated that the incretin effect is severely reduced or even absent in patients with type 2 diabetes (T2DM) [2], even in patients with a considerable insulin secretory capacity. The incretin deficiency is therefore one of the most important pathogenetic factors behind the impaired glucose tolerance in this disease. In early investigations, it became apparent that GIP administration, even in large doses, was unable to restore the incretin effect whether evaluated from insulin responses or glucose turnover [5]. In contrast, infusions of pharmacological amounts of GLP-1 elicited considerable insulin responses that were associated with increases in glucose clearance. Indeed, intravenous infusions of GLP-1 were capable of normalizing completely fasting levels of glucose in patients with long-standing T2DM [6]. This suggested that treatments of type 2 diabetes could be based on restoring the incretin effect [7]. The present review summarizes the current knowledge about the two incretin hormones and the attempts that have been made to use them for the treatment of T2DM. 

## 2. GLP-1

### 2.1. Secretion and Metabolism of Endogenous GLP-1

GLP-1 was first isolated in 1986 from extracts of the porcine intestinal mucosa based on its insulinotropic activity [8]. This peptide hormone is secreted from the so-called L-cells located in the gut epithelium [9,10]. These enteroendocrine cells are distributed evenly throughout the jejunum, ileum and colon [11]. The L-cells express the glucagon gene, resulting in the production of proglucagon, a 160 amino acid peptide [12]. Proglucagon is stored in intracellular granules where it undergoes posttranslational processing and is cleaved by prehormone convertase 1/3 (PC1/3), to yield, amongst other peptides, GLP-1 [13,14]. The L-cells of the ileum and colon (but probably not of the upper small intestine) also produce and secrete peptide YY, which like GLP-1 has anorectic effects [15,16,17]. The two hormones seem to be secreted in parallel but are probably not stored in the same granules [18]. 

Ingestion of nutrients is a strong stimulant for GLP-1 secretion. The L-cell is an “open-type” cell, with an apical cytoplasmic process which is in direct contact with nutrients that arrive in the intestinal lumen [9,19]. Glucose is a strong stimulator of GLP-1 release, and there is a consensus that sodium-dependent glucose transporter 1 (SGLT1) is the dominant luminal glucose sensor in L-cells [20,21,22]. Because of the co Ca^2+^ transport with Na^+^, glucose absorption via this transporter results in a small current which may cause membrane depolarization, followed by voltage-gated Ca^2+^ entry and subsequent exocytosis of GLP-1-containing granules [23,24]. Products of protein degradation are also capable of stimulating GLP-1 release. Peptide transporter 1 (PEPT1) in the luminal and Calcium-sensing receptor (CaSR) in the basolateral membranes of the L-cells are thought to be involved with di-, tri- and oligopeptide stimulated GLP-1 release [25,26]. Peptide transport with PEPT1 involves co-influx of H^+^, again resulting in membrane depolarization and subsequent opening of voltage-gated Ca^2+^ channels [25]. CaSR is a G-protein coupled receptor (GPCR) and activation of this receptor by absorbed amino acids, results in mobilization of intracellular Ca^2+^ [27,28]. Transporter proteins resulting in co-transport of amino acids and ions are also linked to GLP-1 release. Uptake of L-glutamine is for example Na^+^ coupled and results in an elevation of intracellular Ca^2+^ channels [29]. L-glutamine, and other amino acids to a lesser extent, may act to elevate intracellular cAMP levels, another important stimulus to GLP-1 release [29]. As for fat ingestion, several GPCRs have been reported to link fat ingestion to GLP-1 secretion. Activation of GPR120 and GPR40, also called free fatty acid receptor 4 (FFAR4) and FFAR1, respectively, may stimulate GLP-1 release [30,31]. GPR40 is expressed basolaterally on L-cells and senses the presence of FFAs that have already crossed the enterocytes and entered the vascular side in chylomicrons [32]. 2-Monoacyl-glycerol, a product of triglyceride hydrolysis in the gut, binds the G_αs_-coupled GPR119, resulting in cAMP production and subsequent GLP-1 release [33]. The release of bile acids following lipid ingestion has been shown to stimulate GLP-1 secretion via activation of the basolateral TGR5 receptor [34], and secondary bile acids may represent the long-sought stimulus for the secretion of GLP-1 from the colonic L-cells [35].

Other enteroendocrine hormones have been suggested to be involved in the paracrine regulation of GLP-1 secretion. Initially, GIP was thought to stimulate GLP-1 release, but infusion with GIP in healthy individuals does not affect plasma levels of GLP-1 and antagonist-mediated blockage of the GIP receptor did not affect GLP-1 plasma levels [36,37]. Somatostatin (SST) release from neighbouring enteroendocrine D-cells, on the other hand, has shown to reduce GLP-1 release by binding SSTR5 expressed on L-cells [38] and blockade of SSTR5 greatly enhances GLP-1 secretion. 

Extrinsic parasympathetic nerve activity does not seem to play a major role in the regulation of GLP-1 secretion. For example, sham-feeding causing a parasympathetic reaction does not elevate GLP-1 levels [39] and direct stimulation of the vagus is also ineffective [40]. Sympathetic nerves on the other hand seem to inhibit GLP-1 secretion [40] but the exact physiological role of this remains unclear [41]. 

GLP-1 is released and enters the circulation in its intact form, GLP-1(7–36) amide [42]. It is rapidly degraded by the enzyme dipeptidyl peptidase-4 (DPP-4), which cleaves off the first two amino acids at the N-terminus [43]. The resulting metabolite GLP-1(9–36)NH_2_ does no longer possess insulinotropic activity [44]. Both GLP-1(7–36)NH_2_ and GLP-1(9–36)NH_2_ are also degraded by neutral endopeptidase (NEP) 24.11 [45]. As a result of this rapid degradation, less than 10% of secreted GLP-1 reaches its target organs intact [46]. Clearance of both the intact hormone and its metabolites occurs in the kidneys, both through renal extraction and glomerular filtration [47]. 

GLP-1 has also been found to be secreted in neurons in the nucleus tractus solitarii (NTS) in the hindbrain [48]. Central GLP-1 release is stimulated by leptin and gastric distension, but not by peripherally released GLP-1 [49,50].

### 2.2. The GLP-1 Receptor

Intact GLP-1 binds its single known receptor, a class B1 GPCR [51]. The human GLP-1 receptor was first cloned from pancreatic islet cells [52]. Besides expression in the pancreatic α-, β- and δ-cells, the receptor is expressed in several other tissues including the intestine (for example by the somatostatin-expressing D cells [41]), stomach, lung, heart, kidney and several regions of the central nervous system (CNS), such as the hypothalamus and brainstem [53]. Furthermore, GLP-1 receptors are expressed by some of the neurons of the nodose ganglion of the vagus nerve and are thought to mediate neural mediation of GLP-1 effects [54]. 

GLP-1 has a two-domain binding mode to the receptor; firstly, the C-terminus of the peptide binds the large extracellular domain of the receptor, followed by binding of the N-terminus deeper in the transmembrane domain core [55,56]. This binding leads to a conformational change and activation of the receptor, resulting in activation of several downstream signalling cascades [55]. These downstream signalling cascades include elevation of intracellular cAMP and Ca^2+^, as well as ERK1/2 phosphorylation as reviewed recently by Müller et al [57]. Different agonists of the GLP-1 receptor have been found to preferentially activate only one or some of these different pathways, also called biased agonism [56]. This biased agonism is of interest for the development of GLP-1 based therapeutics, as this could direct signalling mechanisms towards a beneficial cellular response to yield optimal therapeutic results [56]. 

## 3. GIP

### 3.1. Secretion and Metabolism of Endogenous GIP

GIP was first isolated in 1970–1975 and was initially named gastric inhibitory peptide [58]. It was, however, later shown that gastric inhibition was only obtained under unphysiological conditions. Its insulinotropic effects, on the other hand, were observed at physiological levels and GIP was therefore renamed to glucose-dependent insulinotropic polypeptide, retaining the acronym [53]. The peptide is secreted from the open-type enteroendocrine K-cells, which are predominantly found in the proximal small intestine, especially in the duodenum. Unlike the L-cells, the density of K-cells in the colon and rectum is very low [59], and some studies even report absence of K-cells in the colon [60]. The 153-amino acid GIP precursor, proGIP, is cleaved by PC1/3 to yield the 42-amino acid bioactive GIP(1–42) [61]. A specific subset of K cells located in the gut and pancreatic α-cells may express PC2, and cleavage of proGIP by this enzyme results in GIP (1–31), which is further converted to GIP(1–30)NH_2_. This truncated form of GIP is biologically active, but circulating levels are very low [62,63]. Like GLP-1, GIP is stored in granules and is rapidly released upon nutrient stimulation [61].

Similar to the L-cells, K-cells express the SGLT-1 transporter and co-absorption of glucose and sodium leads to GIP release associated with elevation of intracellular Ca^2+^ levels [22]. Lipids are likewise potent GIP stimulants. In contrast to what was found for GLP-1 release, absorption of these lipids seems to be essential for GIP release. Knockout of monoacylglycerol acyltransferase 2 (MGAT2) and diacylglycerol acyltransferase 1 (DGAT1), enzymes essential for re-synthesis of triglycerides for release with chylomicrons in the small intestine, results in a significant reduction of GIP release in mice, whereas GLP-1 release is not affected [64]. The basolaterally expressed GPR40 and luminally expressed GPR119 appear to be important for lipid-induced GIP stimulation, whereas GPR120 does not play a major role [65,66]. Protein and amino acids also stimulate GIP secretion and again, CaSR is involved [67]. Neural regulation of GIP secretion has not been uncovered [68,69]. As for hormonal regulation, somatostatin may inhibit GIP release but other regulating factors have not been found [41,70]. 

GIP, like GLP-1, has a short half-life. In humans, plasma half-life of intact GIP was measured to be 7 min [71]. We have recently shown that half-life in mice is even shorter, at approximately 90 s [72]. DPP-4 is responsible for degradation, and cleavage of GIP(1–42) by this enzyme yields GIP(3–42), which has no known biological activity, except for being a very weak antagonist of the GIP receptor [73]. DPP-4 mediated cleavage of GIP(1–30)NH_2_ yields GIP(3–30)NH_2_, which acts as a potent competitive antagonist at the GIP receptor, as further discussed below [74]. NEP24.11 does not appear to degrade GIP under physiological circumstances [75]. 

### 3.2. The GIP Receptor

The GIP receptor is expressed in a variety of tissues, including the pancreas, proximal intestine, adipose tissue, heart, bone and several regions of the CNS [76,77,78], but its function at the majority of these sites is unknown. It is, like the GLP-1 receptor, a member of the class B receptor family. It has a similar structure and is thought to have a similar two-step binding mechanism, although the exact mechanism of receptor activation remains unclear [74]. Several downstream signalling pathways have been proposed for the GIP receptor. Intact GIP(1–42) is a G_αs_-preferring agonist of the GIP receptor, resulting in increased cAMP accumulation in the cell [79]. The GIP receptor is constitutively internalized and recycled in the absence of GIP stimulation, and stimulation with GIP results in desensitization via increased internalization and decreased recycling of the receptor to the cell membrane [79,80]. The GIP-induced internalization of the receptor is dependent on recruitment of β-arrestins upon receptor binding [79]. The receptor does, not only receive signals from plasma, as also internalized receptors on endosomes continue to trigger cAMP production [81]. The GIP system shows more interspecies variations compared to the GLP-1 system. Rodent GIP receptor agonists have shown to be more potent and efficacious than human GIP receptor agonists at both human and rodent GIP receptors [82]. This complicates the translation of in vivo results obtained in rodents to human physiology. 

## 4. The Role of GLP-1 and GIP in Health and under T2DM Conditions

Different GLP-1 and GIP-based treatment strategies for T2DM have been and are currently being developed. To understand the actions of these therapies, we first need to understand the physiological roles of GIP and GLP-1 and how these hormones contribute to the pathophysiology of T2DM. 

### 4.1. Pancreas 

As mentioned above, both GLP-1 and GIP are known as incretin hormones as both hormones are released following ingestion of glucose and potentiate glucose-induced insulin secretion [8,83] (Figure 1). Gasbjerg et al. showed that, upon ingestion of 75 g glucose, the actions of endogenous GIP and GLP-1 are responsible for almost 70% of the resulting insulin response with GIP having the greatest effect [3]. Besides the potentiation of glucose-induced insulin secretion, both incretins have shown to exert proliferative and anti-apoptotic effects on the β-cells [84,85]. It has been proposed that these effects are only achieved by pharmacological levels of the incretins [86]. Furthermore, GLP-1 inhibits the release of glucagon from pancreatic α-cells, whereas administration of GIP at physiological postprandial levels stimulates glucagon release at basal glucose concentrations [87]. The inhibitory effect of GLP-1 on glucagon secretion appears to be mediated via a GLP-1-induced somatostatin release from pancreatic δ-cells, which in turn inhibits glucagon release [88]. GIP does also stimulate somatostatin release, and at elevated glucose levels the stimulatory effects of GIP on glucagon secretion may be obscured by the inhibitory effects exerted by the δ- and perhaps also the β-cells under these conditions [89]. GLP-1, in addition to its endocrine effects, reduces postprandial glucose excursions by inhibiting gastric emptying and intestinal motility through vagal afferent stimulation [90], whereas GIP appears to have no such effects.

As mentioned in the introduction, the incretin effect is significantly impaired in subjects with T2DM, and this is thought to contribute importantly to the postprandial hyperglycemia seen in these patients [2]. GIP secretion increases with obesity, but there does not seem to be a correlation between T2DM and GIP secretion [91]. When measured in large cohorts of patients, average GLP-1 secretion is reduced both in patients with obesity and in those with T2DM. This seems to be a consequence of the condition rather than a primary event [91]. Infusions of GIP and GLP-1 reaching physiological postprandial plasma levels failed to stimulate insulin release in T2DM patients, indicating decreased sensitivity of the β-cells to the incretins [5,92]. Higher infusion rates normalized the β-cell response to GLP-1, whereas GIP only had a small effect on early-phase but no effect on late-phase insulin release [93]. The cause of this reduced sensitivity remains unknown but it may be related to the defective insulin response to glucose since both incretins exclusively influence glucose-stimulated insulin secretion. Improvement of glycemic control, brought about by intensive insulin treatment for 4 weeks partially restored β-cell sensitivity to both incretins (but also to glucose) [92]. The insensitivity of the β-cell to GIP even at high pharmacological roles blunted the excitement for GIP receptor-agonist based therapy of T2DM, as will be further discussed below. 

A-cell function is likewise disrupted by T2DM and patients show elevated glucagon levels, both under fasting and postprandial conditions [94]. Unlike β-cells, α-cells retain their sensitivity to GIP and GLP-1 in T2DM. Unlike in healthy subjects, GIP is capable of stimulating glucagon release at both hypoglycemic, euglycemic and hyperglycemic conditions in these patients [94]. This stimulation of glucagon even at higher glucose levels could be explained by a loss of an insulin-induced inhibitory tone on α cells due to the lack of insulinotropic effects of GIP in patients with T2DM. It has also been suggested that intrinsic changes in the α-cell could be the reason for GIP-induced glucagon release in T2DM patients [95]. Furthermore, elevated plasma amino acids resulting from disturbances in hepatic glucagon signalling in T2DM patients inappropriately stimulate glucagon release in these individuals [96]. GLP-1 seems to retain its suppressing effect on glucagon release under T2DM conditions [97].

### 4.2. CNS

As mentioned above, both the GLP-1 and GIP receptors are expressed in the CNS. GLP-1 affects CNS feeding circuits and reduces food intake through activation of centres related to food-intake, such as the NTS neurons in the hindbrain, the arcuate nucleus (ARC) and the paraventricular nucleus (PVN) in the hypothalamus, as well as the central nucleus of the amygdala [98,99,100]. It is thought that peripherally and centrally secreted GLP-1 have separate effects on the CNS, as blocking central GLP-1 action did not affect the anorexic effect of peripherally administered GLP-1 and vice versa [54]. 

As GLP-1 is rapidly degraded, it has been questioned whether intestinally secreted GLP-1 will directly affect the CNS. It has been suggested that following release from the L-cell, GLP-1 activates GLP-1 receptors transported to the termini in the gut or the hepatic portal bed of vagal, afferent sensory nerve fibres arising from the nodose ganglion [95,101]. These fibres then stimulate neuronal activity in the NTS, which continues to activate the ARC and PVC in the hypothalamus [102]. This is supported by a study where vagotomy in rats reduced the anorexic effects of peripherally administrated GLP-1, accompanied by reduced activation of ARC feeding neurons [103]. In humans, truncal vagotomy also leads to a loss of anorexic effects of GLP-1 infusion [104].

As mentioned previously, GLP-1 is secreted centrally from neurons in the solitary tract, which projects to numerous other brain areas where the GLP-1 receptor is expressed [54,98]. Central administration of GLP-1 was demonstrated to induce an acute anorectic effect in rats, and this effect was blocked by the addition of an antagonist of the GLP-1 receptor, exendin(9–39) [98]. Furthermore, repeated intracerebroventricular (ICV) administration of exendin(9–39) led to a prolonged reduction in food intake and subsequent weight loss [105]. Furthermore, central GLP-1 receptor signalling in the ARC improves glucose tolerance in rats, possibly through stimulation of insulin release from the pancreas as well as inhibition of hepatic glucose production [106]. 

In humans, intravenous infusion with GLP-1 promotes satiety and reduces food intake both in healthy individuals and T2DM patients [97]. A study using functional magnetic resonance imaging (fMRI) showed that GLP-1 affects the central responsiveness to food consumption and thereby is involved in the regulation of food intake; the study also showed that this effect is blunted in T2DM patients [107]. A role for peripheral GLP-1 on food intake in humans is supported by studies performed in subjects that have undergone Roux-en-Y gastric bypass. These individuals show increased postprandial GLP-1 release, which has been linked to the decreased food intake observed in these individuals [108]. 

Much less is known about the effects of GIP in the CNS. The GIP receptor is expressed in feeding centres of the brain, and regions of expression seem to partially overlap with those of the GLP-1 receptor [76,78]. It is, however, unclear if GIP is also centrally secreted [109]. Stimulation of GIP receptor-expressing cells in the hypothalamus leads to an acute reduction of food intake [78] and co-agonism of the GIP and GLP-1 receptor results in a synergistic anorexic effect both in humans and rodents, as will be further discussed below [110,111]. Confusingly, GIP receptor antagonism has also been shown to reduce food intake [112]. Recently, Killion et al showed that, in adipose tissue, prolonged GIP receptor agonism desensitizes receptor activity, hereby mimicking functional GIP receptor antagonism [113]. It has not yet been investigated whether this is the case also in neuronal cells expressing the GIP receptor, and further studies on this topic are warranted. 

### 4.3. Adipose Tissue

A link between GIP and obesity was made just over a decade after its first isolation, as it was suggested that the hormone could promote the deposition of these lipids in adipose tissue [114]. This is further supported by the fact that GIP receptor knockout mice are resistant to diet high fat diet-induced obesity [115]. Furthermore, treatment with GIP receptor antagonists was found to reduce weight gain in rodents and non-human primates [112]. Interestingly, GIP receptor agonism has been reported to show results very similar to antagonist treatment [116]. As mentioned in the previous paragraph, this paradox could be explained through agonist-induced desensitization of the adipocyte GIP receptor [113]. Genome-wide studies have linked the *GIPR* locus to BMI, and diminished receptor activity is associated with decreased BMI [80,116]. Anabolic effects of GIP on adipocytes include stimulation of lipoprotein lipase (LPL) activity and fatty acid uptake [117,118]. GIP is furthermore thought to induce fat accumulation in adipose tissue by increasing substrate availability through stimulating adipose tissue blood flow (ATBF) [119]. However, these effects are abolished by the GIP receptor antagonist GIP(3–30)NH_2_ and blunted in obese subjects, especially in those with impaired glucose tolerance [120]. Not much is known about the adipogenic effects of GIP in T2DM. One study showed that GIP infusion increases subcutaneous adipose tissue lipid uptake, an anabolic effect that would exacerbate obesity and insulin resistance in these patients [121]. 

GLP-1 infusion in healthy individuals increases cardiac output and induces vasodilation both in adipose tissue and particularly in skeletal muscle, even under fasting conditions. This results in increased blood flow to these tissues and has shown in skeletal muscle to enhance insulin-stimulated glucose uptake [122]. It has been reported that GLP-1 receptors are expressed in adipocytes and that receptor signalling induces adipogenesis [123]. To our knowledge, GLP-1 has not been found to have additional adipogenic effects. 

### 4.4. Bone

T2DM patients have an increased bone fracture risk and it has been suggested that the condition leads to impaired bone turnover, resulting in more fragile bone tissue [124]. An enteroendocrine-osseous axis has been proposed to exist, and both GIP and GLP-1 seem to affect bone homeostasis [125]. Reduced GIP receptor activity (due to a functional amino acid substitution) is associated with lower bone mineral density and an increased risk of bone fractures [126]. On the other hand, infusion with GIP both in lean and overweight/obese individuals as well as in T2DM patients is capable of inhibiting bone resorption [125]. Infusion with GLP-1 was also confirmed to increase bone formation in overweight/obese individuals [127]. 

## 5. Therapies for T2DM 

Several incretin-based therapies have been developed or are currently under investigation for treatment of T2DM. These will be discussed in this section. 

### 5.1. Incretin Receptor Agonists

The insulinotropic effects of the first incretin discovered, GIP, initially gave hope for incretin-based T2DM treatment. However, when it was discovered that GIP action is lost in T2DM patients, even at pharmacological doses, this excitement was cooled [128]. Several DPP-4 resistant GIP analogues were developed to study a potential therapeutic application of these analogues in rodent models. DPP-4 resistance and increased peptide stability were achieved through e.g. _D_-Ala^2^ substitution ([_D_-Ala^2^]GIP), PEGylation (GIP[mPEG]) and acylation (GIP(Lys^16^PAL) and GIP(Lys^37^PAL)) [129,130,131]. These compounds showed anti-diabetic effects in obese rodent models, but clinical studies have not been performed. As mentioned above, the pancreatic effects of GLP-1 are maintained T2DM at supraphysiological doses [6], and several GLP-1 receptor agonists have been developed for the treatment of T2DM. Due to the short half-life of GLP-1, analogs had to be developed that would survive DPP-4 and NEP24.11-mediated degradation as well as renal extraction [132]. The peptide exendin-4, isolated from Gila Monster saliva, appeared to be an agonist of the mammalian GLP-1 receptor [133,134]. This peptide may provide a relevant exposure for approximately 5 h following subcutaneous injection of tolerable doses, as it is not sensitive to DPP-4 degradation and is not specifically extracted by the kidneys (merely filtered) [135,136]. A synthetic form of this peptide, exenatide was developed and, following clinical trials, approved for treatment of T2DM in combination with metformin and/or a sulfonylurea as the first GLP-1 receptor agonist [137]. Four years later, a long-acting GLP-1 receptor agonist was approved for once-daily administration – liraglutide [138]. This compound differs from exenatide, as it is composed of acylated mammalian GLP-1 [139]. The attached palmitic acid residue results in albumin binding which prevents renal elimination and induces resistance to DPP-4 degradation [140]. Furthermore, covalent linking of GLP-1 to various, larger molecules has resulted in GLP-1 receptor agonists with further improved half-lives, and these agonists were subsequently marketed [140]. Examples of molecules that only require once-weekly administration are semaglutide [141], which is an optimized liraglutide-like compound, and dulaglutide, which is composed of two stable GLP-1 moieties linked to an immunoglobulin fc fragment [140,142] (Table 1). These compounds are generally given in combination with the oral glucose-lowering drug metformin [143]. Diabetic kidney disease resulting in renal insufficiency is a common complication of T2DM, and renally excreted glucose-lowering therapies such as metformin may therefore be contraindicated for patients. Mammalian GLP-1 receptor agonists are only minimally eliminated via the kidneys and are therefore better tolerated in these patients and have shown beneficial effects on renal outcomes [144,145]. 

In general, the effects of the GLP-1 receptor agonists are similar to those of endogenously secreted GLP-1 (Figure 2). Anti-hyperglycemic effects are achieved through several mechanisms including stimulation of insulin release, inhibition of glucagon release and decreasing gastric emptying rates [146]. Treatment of T2DM with these agents results in improved glycemic control as indicated by reduced hemoglobin A1c (HbA1C) and reduced fasting glucose levels [138,142]. The most significant results on both glycemic control and reduction of body weight are seen with semaglutide [142]. These effects are, as for native GLP-1, glucose-dependent and the risk of treatment-induced hypoglycemia is therefore very low [138]. Furthermore, GLP-1 agonists improved β-cell function and insulin sensitivity [147,148] through different molecular pathways, as reviewed in detail by Yaribeygi et al [146]. Additionally, the GLP-1 RAs induce weight loss in various degrees, presumably through central effects on appetite and food intake [135,138,141,142,149]. This is of great clinical relevance, as weight loss in itself can induce a normalization of hepatic insulin sensitivity as well as improvement of β-cell function, resulting in a normalization of blood glucose levels [150]. In fact, treatment-induced weight loss is the strongest predictor of improved insulin sensitivity [149]. 

In recent years, there has been much interest in the development of GLP-1/GIP co-agonists. Early on it was shown that combined infusions of GIP and GLP-1 have additive insulinotropic effects [151]. The resistance to GIP-induced insulinotropism in T2DM appears to diminish when glucose levels are therapeutically lowered, which provided another rationale for dual GIP and GLP-1 agonism [92]. Tirzepatide, or LY3298176, is a fatty acid modified, 39-amino acid peptide with agonistic properties both at the GIP and GLP-1 receptor [110]. Its structure includes the N-terminal bioactive GIP(1–14) sequence as well as substitutions in the mid-sequence which convey GLP-1 receptor agonism [110,152]. Tirzepatide has shown promising results in clinical trials [110,153]; 26 weeks of treatment with this agent resulted in greater improvement of HbA1C and greater weight reduction compared to dulaglutide [153]. The exact contribution of the GIP component to the efficacy of this compound remains unclear. 

Furthermore, glucagon-based triagonists of the glucagon, GLP-1 and GIP receptors have been developed [157,158]. These triagonists have shown very promising results in both rodent and monkey studies with regards to weight- and glucose-lowering capabilities [158,159], but human data are not yet available. 

### 5.2. DPP-4 Inhibitors

Another strategy for incretin-based T2DM treatment is by inhibition of DPP-4. As this enzyme is responsible for the degradation of both GIP and GLP-1, inhibition of this enzyme yields increased endogenous circulating GIP and GLP-1 levels [160]. Vildagliptin and Sitagliptin are examples of oral DPP-4 inhibitors approved for T2DM treatment. Both have been shown to improve β-cell function in T2DM patients. No weight loss is observed upon treatment with these agents [161]. 

## 6. Conclusions

Incretin based therapy of T2DM has been an unexpected success. Unexpected, because it was anticipated as also indicated by the early observation of the failure of GIP to stimulate insulin secretion in patients with T2DM [128], that their β-cell failure would also compromise any effect of any incretin hormone. With the discovery of GLP-1 and the demonstration that GLP-1, in contrast to GIP, is capable of restoring both beta and alpha cell responsiveness to glucose [7], it became apparent that therapies based on the actions of GLP-1 might be useful for T2DM. DDP-4 inhibitors turned out to be very successful despite their modest effectiveness because of their oral availability and their benign side effect profile. Additional effects of the GLP-1 receptor agonists included their remarkable effect on appetite and food intake, now with the most recent analogs leading to weight losses nearing 20% of body weight, as well as equally unexpected effects on cardiovascular morbidity and survival. Although human studies of acute monotherapy of T2DM with GIP have consistently been disappointing, recent experimental studies have suggested that long-acting GIP receptor agonists may nevertheless influence both body weight and glucose regulation. The effects may be associated with hitherto unrecognized receptor interactions and, most recently, GIP/GLP-1 co-agonists have shown remarkable efficacies with concerning both weight loss and diabetes control. Paradoxically, also GIP receptor antagonists have shown efficacy, at least in combination with GLP-1R agonists. These results point to complex interactions between the co-agonists and the two incretin receptors, and ongoing research is addressing the underlying molecular mechanisms. At any rate, in the years to come, we can expect an increasing use of incretin agonists with highly relevant effects on body weight and marked effectiveness regarding diabetes control and probably also reduction of the risk of cardiovascular complications.

## Figures and Tables

**Figure 1 biology-09-00473-f001:**
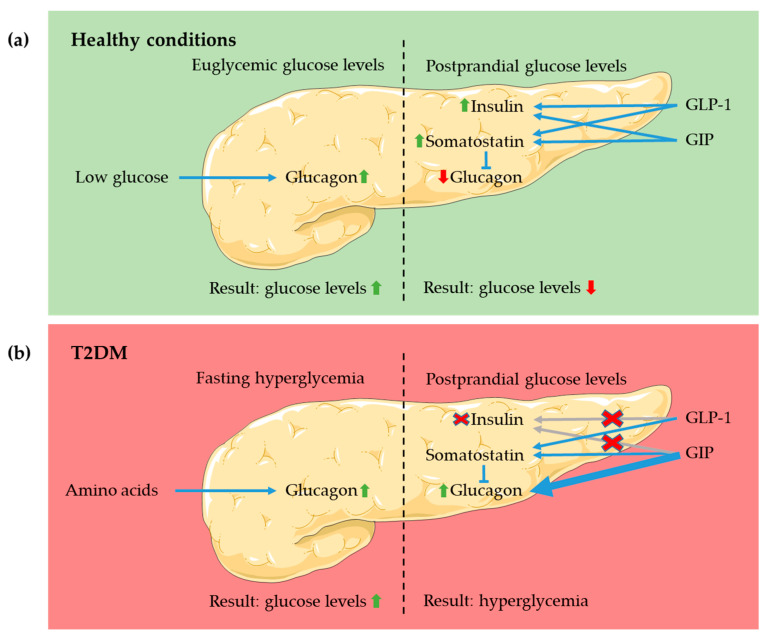
Actions of GLP-1 and GIP in the pancreas. The diagram shows the actions of the incretins under fasting and postprandial glucose levels in (**a**) healthy individuals and (**b**) patients with type 2 diabetes mellitus (T2DM).

**Figure 2 biology-09-00473-f002:**
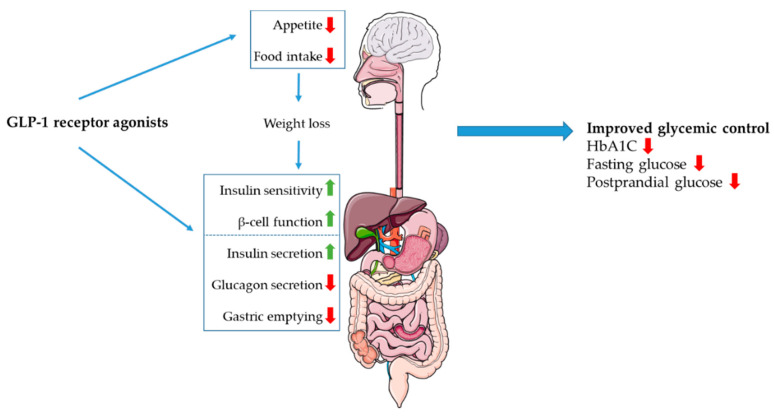
The central and peripheral effects of GLP-1 receptor agonists are shown in this diagram. GLP-1 receptor agonists have shown to reduce food intake and appetite in various degrees. This results in weight loss, which enhances the positive effects of GLP-1 receptor agonists on insulin sensitivity and β-cell function. Furthermore, administration of GLP-1 receptor agonists enhances insulin secretion, whilst inhibiting glucagon secretion and gastric emptying. This results in improved glycemic control in treated patients, as indicated by reduced Hemoglobin A1C (HbA1C) and fasting glucose levels.

**Table 1 biology-09-00473-t001:** Overview of structure, date of FDA approval and dosage frequency of diverse GLP-1 receptor agonists [154,155,156].

Compound	Structure	FDA Approved	Dosage Frequency
Exenatide	Synthetic exendin-4, amino acid substitution at position 2	2005	Twice daily
Lixisenatide	Based on exendin-4, amino acid substitution at position 2. Deletion of proline^36^ and C-terminally addition of a poly-lysine tail	2016	Once daily
Liraglutide	Acylated mammalian GLP-1	2010	Once daily
Dulaglutide	Two stable GLP-1 moieties linked to an immunoglobulin fragment	2014	Once-weekly
Semaglutide	Acylated mammalian GLP-1, amino acid substitution at position 2.	Subcutaneous injection: 2017 Oral administration: 2019	Subcutaneous injection: once weekly Oral administration: once daily

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
