# Peer review of "Incretin Hormones and Type 2 Diabetes—Mechanistic Insights and Therapeutic Approaches"

_biology, 2020, doi:10.3390/biology9120473_

Round 1
Reviewer 1 Report
The manuscript by Boer and Holst is very topical in the area of diabetes research. My overall impression is that the manuscript is generally well written and most of the key scientific literature is included. There are some errors (grammatical) that can be fixed up easily with thorough proof-reading. The authors sometimes use words like 'drastically' and I would suggest moderating these. Several errors were noted with some of the references so I urge the authors to cross-check these. Regarding the scientific content there are two observations that pertain to the therapeutic sections - basically these could be enhanced. Firstly, the authors should include key literature on early development of GIP and GLP-1 analogues, at least acknowledge this work and cite appropriate references. Secondly, the latter sections around dual/triple or combination therapies is brief and could be enhanced with additional detail. If word count is a problem I would suggest cutting back on some of the earlier physiology sections.
Reviewer 2 Report
The authors wrote a comprehensive and timely review that summarizes current knowledge in the field on the mechanisms of action on two incretin hormones and the development of incretin-based therapies for treatment of type 2 diabetes. This up to date review provides to the readers a new look at of mechanisms of action of these two hormones across different organs and tissue, which is strength. I do not have any other comments or recommendations to the authors. A very good job.
Reviewer 3 Report
This review aims to evaluate current knowledge of the two incretin hormones, GLP-1 and GIP, and the development of incretin-based therapies for treatment of type 2 diabetes.
The authors concluded that we expect an increasing use of incretin agonists with highly relevant effects on body weight and marked effectiveness regarding diabetes control and probably also reduction of the risk of cardiovascular complications.
This article is well written and of clinical interest.
However, several issues should be improved before the consideration for publication.
Major comments
1 Patients with diabetes predispose to result in renal failure. However, most of the oral drugs are unsuitable for the patients with renal failure. Many physicians would like to know the applicability of GLP-1 receptor agonists in diabetic patients with advancing renal failure, particularly compared with other oral medications.
2 In line 375, although the authors mentioned “now with the most recent analogs leading to weight losses nearing 20% of body weight,”, I wonder 20% may be overstatement. Please add corresponding literatures.
3 A summary table of GLP-1 Ras: Dulaglutide Exenatide Liraglutide, and Lixisenatide is informative.
Concise descriptions of differences including benefits, clearance rates from the kidney, and disadvantages are helpful.
4 The effects of GLP-1 on gluconeogenesis in the liver/kidney and the metabolism in skeletal muscle are emerging topic. Please mention these issues.
5 A combination therapy of GLP-1 with metformin has been frequently conducted for the treatment of T2DM.
Author Response
- Patients with diabetes predispose to result in renal failure. However, most of the oral drugs are unsuitable for the patients with renal failure. Many physicians would like to know the applicability of GLP-1 receptor agonists in diabetic patients with advancing renal failure, particularly compared with other oral medications. We acknowledge the importance of this issue and have included a brief discussion on this on Page 9, Line 346-351.
- In line 375, although the authors mentioned “now with the most recent analogs leading to weight losses nearing 20% of body weight,”, I wonder 20% may be overstatement. Please add corresponding literatures.
We have added corresponding references to this text (Page 10, Line 398). In the three STEP trials that form the basis for the new drug application for semaglutide to the FDA as of December 7, weight losses were around 18% (https://ml-eu.globenewswire.com/Resource/Download/d8a1f4e0-82a7-4d5f-bb70-f6b96a9ef77d and https://www.novonordisk.com/content/nncorp/global/en/news-and-media/news-and-ir-materials/news-details.html?id=278); similar findings were made in the phase two trial for semaglutide (O'Neil PM, Birkenfeld AL, McGowan B, Mosenzon O, Pedersen SD, Wharton S, Carson CG, Jepsen CH, Kabisch M, Wilding JPH. Efficacy and safety of semaglutide compared with liraglutide and placebo for weight loss in patients with obesity: a randomised, double-blind, placebo and active controlled, dose-ranging, phase 2 trial. Lancet. 2018 Aug 25;392(10148):637-649. doi: 10.1016/S0140-6736(18)31773-2. Epub 2018 Aug 16. PMID: 30122305.). Currently, combinations with cagrilintide (AM833, a long-acting amylin analog) and semaglutide (Cagrisema) seem to give even larger weight losses. - A summary table of GLP-1 Ras: Dulaglutide Exenatide Liraglutide, and Lixisenatide is informative. Concise descriptions of differences including benefits, clearance rates from the kidney, and disadvantages are helpful.
We agree with the referee that a table is informative in this case, and we have added a table depicting compound name, structure, FDA approval year and dosage frequency (Table 1, page 9). We have chosen to not to include further details regarding the benefits and disadvantages pertaining to these different compounds; the diverse clinical studies performed are very complex and are performed under different conditions. To fully capture results obtained with these studies, we would require a much more detailed description of the studies, which we feel lies outside the scope of our current review. - The effects of GLP-1 on gluconeogenesis in the liver/kidney and the metabolism in skeletal muscle are emerging topic. Please mention these issues.
We believe that the main effect of GLP-1 on metabolism in skeletal muscle is to increase blood flow in this tissue and to enhance insulin-mediated glucose uptake. This has been described on Page 7, Line 303-305. We are convinced that there is no expression of GLP-1 receptors on hepatocytes and the well documented receptors in the kidneys are located to the afferent arterioles. Therefore, it is hard to imagine a direct role for GLP-1 in these tissues. - A combination therapy of GLP-1 with metformin has been frequently conducted for the treatment of T2DM. We acknowledge the important role of metformin within T2DM treatment and have added this note to the manuscript: Page 9, Line 346-347. The GLP-1 RAs have also been studied in monotherapy e. g. Sorli C, Harashima SI, Tsoukas GM, Unger J, Karsbøl JD, Hansen T, Bain SC. Efficacy and safety of once-weekly semaglutide monotherapy versus placebo in patients with type 2 diabetes (SUSTAIN 1): a double-blind, randomised, placebo-controlled, parallel-group, multinational, multicentre phase 3a trial. Lancet Diabetes Endocrinol. 2017 Apr;5(4):251-260. doi: 10.1016/S2213-8587(17)30013-X. Epub 2017 Jan 17. PMID: 28110911.
Round 2
Reviewer 3 Report
The manuscript has been improved according to the comments.
Thank you for interesting paper.